# Hybrid and Plant-Based Burgers: Trends, Challenges, and Physicochemical and Sensory Qualities

**DOI:** 10.3390/foods13233855

**Published:** 2024-11-28

**Authors:** Ziane da Conceição das Mercês, Natalia Maldaner Salvadori, Sabrina Melo Evangelista, Tatiana Barbieri Cochlar, Alessandro de Oliveira Rios, Viviani Ruffo de Oliveira

**Affiliations:** 1Postgraduate Program in Food, Nutrition and Health, Federal University of Rio Grande do Sul (UFRGS), Porto Alegre 90610-264, RS, Brazil; zianemerces@gmail.com (Z.d.C.d.M.); natisalvadori18@gmail.com (N.M.S.); sabrina.evangelista@ufrgs.br (S.M.E.); tatianabarbieri2010@hotmail.com (T.B.C.); 2Institute of Food Science and Technology, Federal University of Rio Grande do Sul (UFRGS), Porto Alegre 91509-900, RS, Brazil; alessandro.rios@ufrgs.br; 3Department of Nutrition, Federal University of Rio Grande do Sul (UFRGS), Porto Alegre 90610-264, RS, Brazil

**Keywords:** patties, plant-based burgers, meat analogs, hybrid burgers

## Abstract

Burgers have become a staple of global cuisine and can have several different versions and combinations. For example, hybrid burgers have a percentage of animal protein in their formulation, while plant-based burgers contain 100% plant-based proteins. Therefore, the aim of this study was to investigate the emerging trends and challenges in the formulation of hybrid and plant-based burgers, with an emphasis on new ingredients and the evaluation of their physical, chemical, and sensory properties. An integrative literature review on alternative burgers to meat ones was carried out, focusing on hybrid products (meat + plant-based) and fully plant-based burgers. The studies analyzed show that plant-based and hybrid burgers can be developed with different protein sources, such as soybeans, white beans, textured peas, pseudocereals, and cashew nuts, with good nutritional and sensory characteristics. While hybrid burgers combine meat and plant-based proteins to reduce saturated fats, plant-based burgers show equal promise, with a high protein and fiber content, a lower fat content, and good sensory acceptance. However, despite the market potential of these products, there are challenges to be overcome, among which are their texture and flavor, which are essential characteristics of animal-meat burgers. Another point to take into account is the diversity of preferences among consumers with different beliefs or eating styles: vegans, for example, do not prefer a product that is very similar to meat, unlike flexitarians, who seek products which are similar to animal meat in all attributes.

## 1. Introduction

Burgers are described as convenience food, or fast food, which consists of a meat product, usually made from beef, and this meat is placed between two halves of a bun. Hybrid burgers combine animal-based ingredients with vegetables, such as legumes or grains, to reduce the meat content and improve the nutritional profile. Plant-based analog burgers, on the other hand, are plant-based products developed to mimic the flavor, texture, and protein value of meat, aiming to offer a sustainable alternative free of animal components [1].

In 2022, The Good Food Institute [2] reported that the global retail sales of plant-based meat substitute products amounted to USD 6.1 billion. According to the Grand View Research report [3], the global vegan food market is expected to reach USD 37.5 billion in 2030.

Another segment that has been growing is the market for plant-based meat alternatives, also known as meat analogs [4]. Plant-based products have been considered meat alternatives and have experienced a significant increase in new product launches worldwide in recent years, especially in categories such as burgers [5], ground beef [6], sausages [7], and meat balls [8]. This rapid growth in product supply and availability is creating business opportunities for the plant-based meat substitute industry [9]. From 2018 to 2020, the European plant-based meat sector’s market share grew by 68%, and it is expected to continue to grow in the coming years [10].

Meat alternatives are substitutes that propose to mimic some of the sensory attributes, like appearance, flavor, and texture, of animal meat, and they have received different nomenclatures such as plant-based meat [11], meat alternatives [12], vegetarian meat [11], vegan meat [13], meat analogs [14], plant-based protein [11,15], meat substitutes [12], fake meat [12], and hybrid burgers [1]. These meat analogs are also defined as non-traditional food products that need to have sensory, chemical, and technological qualities that resemble meat [5]. These meat substitutes aim to provide more food options for consumers [7,16,17].

From this perspective, burgers are among the foods that stand out for their versatility [18]. Hybrid burgers, for example, combine animal-based meat and plant-based proteins to cater to a diverse audience [18,19,20]. Another aim of hybrid products is to reduce the consumption of animal meat by incorporating more ingredients of plant origin. This combination, present in hybrid foods, not only decreases the amount of meat but also contributes to reducing the environmental impact while supporting many of the sensory characteristics of traditional meat-based products [1]. Meanwhile, meat analog burgers are formulated with plant and/or mushroom protein sources without the addition of animal meat [1].

The increasing availability of plant-based burgers, developed to address environmental, health, and ethical issues related to animal production, represents a shift in the alternative protein market [9]. However, these alternatives are often perceived as less satisfying and sensorily appealing compared to traditional burgers [9].

From this perspective, this study looked to identify new ingredients used as a basis for the formulation of hybrid and plant-based burgers while also investigating other possible raw materials for innovative formulations. This approach is essential to diversify sustainable protein sources and enhance the sensory properties of products, meeting the growing demand for healthy and environmentally responsible food alternatives.

Therefore, the aim of this study was to investigate emerging trends and challenges in the formulation of hybrid and plant-based burgers, with an emphasis on new ingredients and the evaluation of their physicochemical and sensory properties.

## 2. Methodology

An integrative literature review was carried out on burgers that could be consumed as alternatives to meat, focusing on hybrid products (meat + plant-based) and fully plant-based burgers.

The search for available scientific articles was carried out on the following databases: PubMed, Scientific Electronic Library Online (Scielo), Science Direct, Scopus, Food Science and Technology Abstracts (FSTA), Latin American and Caribbean Health Science Literature, and Wiley Library. Appropriate terms were used for each database according to the Food Science Source Thesaurus, Medical Subject Headings (MeSH), Embase Subject Headings (EmTree), and Health Sciences Descriptors (DeCS), using a combination of keywords in English—“hybrid burger”, “plant-based burger”, “meat analog”, “meat-like burger”, “hamburger”, and “quality of burger”—and the Boolean operator AND.

The inclusion criteria for the selection of studies were the following: (1) original research articles, systematic reviews, and review articles published between 2014 and 2024; (2) documents published in peer-reviewed journals; and (3) studies focused on innovative approaches to the development of hybrid or plant-based burgers.

The following exclusion criteria were used: (1) lack of compatibility with the main theme; (2) studies involving animals in experiments; (3) case reports, conference abstracts, patents, and letters to the editor; (4) non-scientific sources such as websites, editorials, and blogs; and (5) articles of low methodological quality.

All the selected studies were organized and managed using Mendeley software (version 2.116.1), eliminating duplicate articles. The articles were identified, and a detailed screening was carried out to confirm their relevance in relation to the research topic. They were then organized according to the established eligibility criteria, read in full, and included in this study, as shown in Figure 1.

## 3. Meat Production, Diet Changes, and New Food Possibilities

World meat production has more than doubled since 1961, accompanied by increasing environmental impacts [21]. Transitioning from an animal-based diet to a plant-based diet can be a strategy to promote healthier eating and footprint reduction [21].

The market for plant-based products was valued at USD 8.9 billion in 2019 and is predicted to generate revenues of USD 14.3 billion by 2025, while animal meat substitutes or meat analogs will exceed USD 24 billion [22,23]. In 2022, The Good Food Institute [2] reported that global retail sales of plant-based meat substitutes amounted to USD 6.1 billion. According to the Grand View Research report [3], the global vegan food market is expected to reach USD 37.5 billion by 2030. The variety of meat analogs available nowadays is huge, covering diverse types of products, recipes, and characteristics. The media declared 2019 the “year of the plant-based burger”, highlighting that plant-based products are gaining the most recognition and are constantly improving [24].

Among the many varieties of plant-based burgers already described in the literature, there are formulations containing cereals and pseudocereals [25], legumes [13,26,27], nuts [28], vegetables [27,28,29], fruits [28], edible mushrooms [30,31], and algae [27,32].

Figure 2 illustrates the process of developing new food products from sustainable ingredients, highlighting the importance of planning and testing formulations that have nutritional, technological, and sensory potential and the quality to meet the demands of the greatest number of consumers.

In the twenty-first century, socio-economic and dietary changes have caused a nutritional transition and influenced the uptake of new dietary choices, such as vegetarianism, veganism, and flexitarianism, in response to population growth and new food demands [24]. It is estimated that millions of people have adopted vegetarian diets, which exclude meat, or vegan diets, which avoid all animal derivatives [32]. These groups have been expanding despite being minorities in wealthy nations [33,34]. An example of this increase can be seen in the annual “Veganuary” campaign, which promotes the adoption of a vegan diet during the month of January. In 2022, registrations reached a record high, with more than 700 thousand people from almost every country in the world taking part, showing an upward trajectory from the 3300 participants in 2014 to 168,500 in 2018 and 582,000 in 2021 [35].

Therefore, the food industry, realizing the changes in selective eating routines and the increase in the population, has been working on new products, and among the new foods are plant-based meat analogs [36]. These new foods are driving the market for vegetarian, vegan, and flexitarian products, reflecting the preferences of these consumers and creating economic opportunities [37].

However, there are some challenges with plant-based diets regarding nutritional quality, textural properties, and consumer acceptance [38,39,40]. A lack of familiarity with these products can also lead to negative consumer expectations and lower eventual acceptability [38].

In this sense, plant-based meat alternatives are being developed to meet consumers’ demand and a sustainable future food supply, and the market has grown exponentially in recent years [41]. Thus, alternative plant protein products such as plant-based burgers, which have been around for many decades and were once considered a niche sector of the food industry, are experiencing rapid market expansion [39].

### 3.1. Hybrid Burgers

Interest in flexitarian diets has been growing and is reflected in the increasing number of consumers adopting meat-free days, opting for plant-based alternatives and reducing meat consumption for various reasons, such as health, environmental sustainability, and animal welfare [41]. Flexitarian diets include more products based on plant-based raw materials or hybrid products containing meat and plant-based ingredients [42]. The food industry, noticing the changes in selective eating routines, has been working on new products, and among the new foods are hybrid burgers [42].

Hybrid burgers present both benefits and challenges to the food sector. The incorporation of raw plant materials in these products influences their sensory and technological properties, including appearance, texture, and taste, as well as provides higher fiber and mineral content and reduces the caloric value of the product [43]. Hybrid meat products can fill the gap between products aimed at consumers who want to reduce their meat consumption without sacrificing the taste, convenience, and familiarity of traditional meat products [42].

Currently, the definition of hybrid meat products is not yet clearly established. These products are often characterized by the inclusion of plant-based ingredients, such as legumes, cereals, fruits, algae, and mushrooms, in proportions ranging from 25% to 50% [42]. These ingredients are not only used as additives but also incorporated for their technological, nutritional, and sensory properties, which enrich the product [43,44].

Petrat-Melin and Dam [1] recommended a combination of 50% animal protein (beef, pork, and poultry) and 50% plant-based proteins, while Profeta et al. [45] suggested replacing a smaller fraction of the meat product, 20–50% plant proteins.

Profeta et al. [45] reported that meat hybrids are a possible alternative for consumers who are not interested in totally vegan or vegetarian alternatives to meat. Thus, hybrid burgers could represent an affordable option to facilitate this group’s transition to a healthy and sustainable diet.

Ziegler et al. [46] made ten beef burger options with unconventional plant-based edible ingredients such as yacon roots (*Smallanthus sonchifolius*), moringa (*Moringa oleifera*), and ora-pro-nóbis (*Pereskia aculeata* Mill.) roots and evaluated the physical, chemical, and sensory properties. The formulations included a soy protein (conventional burger) and nine formulations with variations of 2%, 4%, and 6% flour from these plants. The means and standard deviation of the analysis showed that the proximate composition (dry basis) of *yacon*, *moringa*, and *ora-pro-nobis* flours included 71.77, 13.83, and 1.71% non-fiber carbohydrates; 21.03, 48.21, and 55.39% dietary fibers; 4.67, 7.75, and 16.34% ashes; 1.96, 24.79, and 22.41% proteins; and 0.57, 4.54, and 3.27% fat, respectively. 

In the sensory test, Ziegler et al. [46] presented data showing the differences between burgers prepared with different concentrations of *yacon, moringa*, and *ora-pro-nobis* flours and the conventional burger. It can be seen that when 2% *Yacon flour was used*, the values varied between 2 and 3 for parameters such as color, taste, aroma, and texture, which showed that the burgers were classified as “slightly different” from the control. As the concentration of flour increased, the color, taste, and texture scores increased significantly (*p* ≤ 0.05), and with the addition of 6% *yacon* flour, these scores were 3.31, 5.18, and 3.68, respectively.

For the authors, one of the great challenges of food technology is the production of foods with nutritional, functional, and bioactive properties without causing major changes in sensory characteristics. In this context, it was observed that the formulations closest to the conventional burger were those produced with 2% and 4% *yacon* flour, which may be well accepted by consumers who are looking for healthier foods [46].

Atitallah et al. [47] evaluated canned fish burgers with the addition of algae, and the results showed that the addition of microalgae significantly improved the physico-chemical composition and sensory acceptability of the fish products without altering their microbiological quality. Among the results, the authors presented data showing that the incorporation of microalgae in hybrid fish burgers had no effect on pH values. In addition, there were no statistically significant differences in protein and fat contents between the control and microalgae-fortified samples (*p* ≥ 0.05). The authors observed that microalgae were effective natural additives for canned fish burger (a product derived from sliced barbel fish meat) and that this enrichment provided a natural source of bioactive substances (chlorophylls, carotenoids, and phycocyanin) to the product.

Chaves et al. [48] made burgers in which beef was partially replaced by textured soy protein (TSP) and mixed plant flour, carried out yield analyses after cooking, and performed color determination and sensory evaluation. The results showed that it is feasible to develop a burger with the addition of mixed plant flours, as the sensory evaluation showed an acceptability index exceeding 70% for all attributes evaluated. Furthermore, the averages were higher compared to the control formulation, especially in relation to the attributes taste and texture.

In the study conducted by Barker and McSweeney [49], yellow pea flour was incorporated in different proportions (0%, 10%, 20%, 30%, and 40%) into a chicken burger patty. The sensory evaluation included 69 assessors using hedonic scales and the Check-All-That-Apply (CATA) methodology. The addition of yellow pea flour reduced preference for the burgers, except in the 10% formulation. As described by the authors, burgers made with higher amounts of yellow peas were associated with off-flavors (beans and nuts) that were significantly different (*p* < 0.05) from the control. Their study also reported that yellow peas contributed to a dry texture that was not appreciated by consumers.

Rosa and Lobato [50] made artisanal burgers using cashews (*Anacardium occidentale* L.): a cashew juice-based patty (F1 = 15%) and another with cashew fiber (F2 = 9.5%), both combined with ground beef (60–75%, respectively) and textured protein (5–6%). According to the analysis data, the color was associated averages of 6.83 (F1) and 6.72 (F2); taste, 6.97 (F1) and 6.87 (F2); and texture, 6.35 (F1) and 6.56 (F2). Finally, the overall averages were 7.27 (F1) and 7.12 (F2). Thus, there was a slight preference for burgers made with cashew juice over those made with fiber. Their study mentioned the relevance of cashew as a raw material in burger formulation considering its nutritional properties. In the sensory analysis, the purchasing intention for the burgers with cashew juice and fiber was questioned, and 66.66% of the 48 assessors reported that they would “Probably buy” or “Certainly buy” the burger made with cashew juice (F1).

In the study by Adeniyi et al. [51], they presented an experimental study designed to compare the nutritional, physical, and sensory properties of soy burgers with those of conventional beef and chicken burgers. The protein content (dry weight basis) for beef burgers (BBs), chicken burgers (CBs), and soy burgers (SBs) was 57.87, 50.36, and 53.89%, respectively. In a similar sequence, the fat content was 10.02 (BB), 10.15 (CB), and 8.58 (SB). The SBs showed significantly higher amounts of crude fiber, riboflavin, and niacin than the other two samples, but its cholesterol content was not significant. Chicken burgers had the highest firmness (17.71 N), followed by beef (16.54 N) and soybean (15.40 N) burgers, with statistically significant differences between the groups (*p* ≤ 0.05). Regarding chewiness and cohesiveness, there were no significant differences between the burgers, showing similar results.

Santos et al. [52] made chicken burgers with the addition of green banana biomass and passion fruit epicarp. The production of burgers with 10% banana biomass may represent an alternative to enriching and maintaining their physicochemical parameters, in addition to their technological properties. Improving texture was indicated as the most important benefit for improving sensory parameters such as firmness, gumminess, and chewiness. The sample with green banana produced the best results; however, the proposed use of a co-product such as passion fruit epicarp may also be a great solution to reducing waste and costs in kitchens and the food industry.

Bahmanyar et al. [53] studied the effects of replacing soy protein and bread flour with quinoa and buckwheat flour in the formulation of functional beef burgers. The results revealed significant differences in the chemical composition, pH, antioxidant activity, and total phenolic content between the different flours investigated. Soy protein powder had the highest protein content (49.44 g/100 g), followed by quinoa flour (14.12 g/100 g), while bread flour (10.30 g/100 g) and buckwheat flour (9.8 g/100 g) showed lower values. In terms of fat content, the burger with quinoa flour stood out, with the highest concentration (7.28 g/100 g), while bread flour had the lowest content (0.46 g/100 g).

Rabadán et al. [54] reported that completely replacing beef fat with tiger nut oil emulsion (*Cyperus esculentus* L.) in burgers resulted in a healthier product, with low total and saturated fats and being rich in unsaturated fatty acids, especially oleic acid. In agreement with these results, the study by Carvalho Barros et al. [55] highlighted the benefits of replacing beef fat with a tiger nut oil emulsion in beef burgers, resulting in a healthier meat product.

Dasiewicz et al. [42] determined the influence of the proportions of the meat and plant-based parts; according to the authors, hybrid and plant-based burgers had similar water, protein, and fat contents and cooking yields. Compared to the beef burger, however, one of the possible obstacles in making hybrid burgers containing meat and plant-based may be their soft texture, which determines the perception and final acceptance of the products by consumers.

Some authors have also discussed hybrid burgers and their combination with vegetables, fruits, legumes, cereals, mushrooms, algae, and nuts, showing that these products are relevant to society, as shown in Table 1.

#### Hybrid Burgers: A Path Towards Plant-Based Burgers?

Hybrid products, such as burgers, nuggets, and sausages, have provided not only advantages but also challenges for the food industry. The addition of raw plant materials can affect the technological and sensory properties of hybrid products, particularly their appearance, texture, and taste [42].

At first, plant-based meat analogs were developed to interest omnivores and flexitarians in particular, as these consumers appreciate the similarity of these products to meat, both in terms of texture and flavor [15,59].

Plant-based meat analogs are an emerging field in the current literature, with important knowledge gaps to be filled [60,61]. In this context, hybrid meat products—which replace part of the meat with more sustainable protein sources—could be a good option for covering the gap between meat foods and meat-free foods, offering convenience and allowing consumers to maintain their traditional eating habits [43]. Moreover, the combination of meat and plant ingredients in hybrid foods aims to reduce meat consumption and climate impacts while maintaining the sensory characteristics of pure meat [1]. Although earlier efforts to commercialize these products have been unsuccessful [43], the partial replacement of meat with plant-based alternatives has recently gained greater acceptance with the rise of plant-based meat analogs.

A brief explanation of what the term “plant-based meat” means was explained by Lee et al. [62], who reported that plant-based meat refers to textured foods produced from plant proteins, with the purpose of mimicking or replacing meat. The global market for plant-based proteins is projected to grow at a compound annual growth rate of 7.2%, reaching USD 15.6–21.23 billion by the year 2026 [7,15,62].

Plant-based protein burgers have become increasingly popular among consumers driven by concerns about food safety, environmental protection and sustainability [63].

According to Coucke et al. [14], the evolution of meat analogs launched on the market is constantly growing. These advances are reflected in food processing technology, which allows meat analogs to be made using sources other than plant-based ingredients or microproteins from plant origin.

This statement agrees with the study by Mercês et al. [64], which identified several challenges in formulating plant-based meat analogs, especially in mimicking the sensory attributes of meat. Notably, preferences vary among consumers, with vegetarians and vegans looking for plant-based alternatives with distinct characteristics of meat but omnivorous and flexitarian consumers preferring options that come as close as possible to the taste of meat. In some situations, these consumers are reluctant to purchase plant-based protein alternatives, fearing that these products may not meet their sensory expectations [65,66].

Meat analogs, with the sensory attributes of meat, can represent a viable solution for satisfying customers’ dietary preferences [12,63,67]. The acceptance of meat analog burgers can have a positive impact on society when considering dietary and environmental changes. Several factors, such as color, texture, price, as well as social, economic, and nutritional aspects, can influence consumers’ choice of whether to include meat substitutes in their diet [11,12,63,68]. The growing global effort to reduce meat consumption means that this initiative is gaining more and more support [69].

Grasso [70] mentioned that it is difficult to know or predict what the future of hybrid meat products will hold. It is possible that the taste and aroma of 100% plant-based options will continue to improve, with growing global interest and continued research and investment in the subject. Therefore, a clean shift from meat to non-meat diets will take place, and hybrid meat products will become superfluous, giving full space to plant-based meat analogs.

Grasso [70] emphasized that hybrid meat products can be an easy way to reduce meat consumption without compromising on taste or having to develop new cooking skills or habits, as well as having the benefits of adding plant ingredients and contributing to the nutritional transition to meat-free foods.

### 3.2. Plant-Based Proteins Used in Burger Making

#### 3.2.1. Legumes in Burgers

Belonging to the botanical family Fabacae or Leguminosae, legumes comprise a diverse group, which include dried beans, broad beans, peas, chickpeas, cowpeas, lentils, lupins, and soybeans [71,72,73]. These foods are a viable alternative in the production of meat analogs due to their protein capacity and additional health benefits, as well as their lower levels of saturated fat than proteins of animal origin and their importance as sources of fiber [74,75].

Soy protein has already been used as a popular ingredient in meat analogs, such as tofu and tempeh—the latter is a soy meat analog created by fermenting soybeans or other legumes such as peas, beans, or chickpeas with strains of the filamentous fungus *Rhizopus* sp. [16]. Textured vegetable protein was introduced as an alternative to meat in the middle of the 20th century. Furthermore, dry textured plant protein was the first meat analog obtained from defatted and extruded soybean meal, with a high content of soy proteins or wheat gluten [16]. Another interesting example was the incorporation of isolated proteins of legumes, such as peas or beans, into meat alternatives, which has been considered a promising strategy [59].

Vital et al. [76] made a white bean tempeh burger and compared it with a soy tempeh burger. In their sensory analysis, the assessors showed a higher preference (68.29%) for the soy burger than for the white bean burger (47.56%). In relation to nutritional quality, the soy burger had higher levels of fat, calories, and protein, while the white bean burger had significantly lower levels of fat and calories, but also lower amounts of protein. These results indicate the superiority of the soy burger in terms of acceptability and protein composition, despite having a higher calorie intake, as shown in Table 2.

Herawati, Kamsiati, and Widowati [29] described the effects of combining the porang tuber (*Amorphophallus muelleri*) with tempeh beans on the characteristics of vegetarian burgers. The results indicated significant variations in the color of the burgers before and after cooking, with a reduction in brightness levels and an increase in yellow color as the proportion of porang tuber increased. The combination also influenced cooking loss and yield: the 75:25% porang/tempeh tuber had a cooking loss of 23.27% and a cooking yield of 76.73% compared to the 25:75% porang/tempeh tuber.

Legume proteins have been shown to be effective in modulating plant burgers, preserving the texture properties without compromising the taste of the product [26]. The protein source, soy, or processed legumes proteins in plant-based burgers with meat analogs made with high-moisture extrusion did not significantly affect the preference of the 80 consumers who evaluated appearance and taste. Despite some inferior textural properties, processed legume proteins have a unique combination of attributes, which makes them a promising alternative for the production of plant-based burgers.

Sousa et al. [13] carried out a study to compare the in vitro digestibility and the essential amino acid ratio between vegan and meat burger before and after grilling. In the raw state, the digestibility of the plant protein sources tested was approximately 85%. However, the digestibility of the beef burger was almost 100%, significantly higher than of the plant protein sources (*p* < 0.005). In this sense, the results show that the alternative protein sources tested proved to be good alternatives to meat due to their high digestibility values.

In pea-textured protein burgers with the addition of alfalfa (*Medicago sativa*), spinach (*Spinacia oleracea*), and chlorella (*Chlorella vulgaris*), the effect of adding these ingredients was evaluated in the powdered and textured versions, considering the physicochemical properties and extrusion parameters [27]. The results showed that an enrichment with textured pea protein made it possible to make a product similar to a traditional beef burger due to the fibrous texture obtained after the extrusion process, which was more similar to the texture of meat and not as grainy as the samples produced with powder. The same authors also added that the enriched formulations improved tenderness, juiciness, and chewiness but decreased adhesiveness. Burgers made with textured pea protein had a texture similar to meat, with greater hardness, juiciness, and chewiness and lower graininess and adhesiveness. The use of *Chlorella vulgaris* reduced the legumes’ aroma, while burgers with textured pea protein intensified the overall spicy aroma [27].

#### 3.2.2. Cereals and Pseudocereals in Burgers

Cereals such as rice, barley, or oats are measured according to the degree of processing and can be classified as seeds, flours, brans, or flakes [7]. Burgers containing cereals and/or pseudocereals have also been studied for their nutritional and sensory properties [46].

Chilon-Llico et al. [25] pointed out that burgers made with lupins, quinoa, and corn replacing meat can be a source of complementary protein with high nutritional value. Despite the limited research on vegetarian blends and protein quality, sensory results indicated that these burgers are described as easy to cut, soft, good, healthy, legume-flavored, tasty, and light brown in color.

Summo et al. [75] made pre-cooked legume-based burgers using three different formulations (L1, L2, and L3), in which the legumes were in equal proportions (lentils, peas, and beans); the formulations also included cereals in varying proportions (barley and/or corn). The results showed that the protein content of the burgers varied between 8.57% and 9.15% on a wet basis. More than 12% of the energy value of burgers was provided by proteins, with amounts of 12.39% in LB1, 13.04% in LB2, and 12.78% in LB3. The total fiber content of the burgers ranged from 5.58% to 7.88% on a wet basis, with significantly higher values in LB2. Oleic acid was the predominant fatty acid in all formulations, with amounts ranging from 57.59% to 59.34%. Monounsaturated fatty acids represented about 61% of the total fatty acids.

#### 3.2.3. Burgers with Added Nuts and Fruit

Lima et al. [28] evaluated a by-product of cashew nut processing to make a protein concentrate and characterized it for use in a plant-based burger formulation to replace soy protein. The burgers made with the cashew nut protein concentrate had the following composition: 1.5% lipids, 7.1% proteins, 18.7% carbohydrates, and a total energy value of 115.4 kcal/100 g. On the other hand, the soy protein-based burgers had 1.3% lipids, 6.7% proteins, 19.6% carbohydrates, and a total energy value of 118.0 kcal/100 g. Cashew nut protein concentrate with 58.6% protein content can be produced via isoelectric precipitation in a pH range of 4.0 to 4.5. This cashew nut protein concentrate showed high water and oil absorption capacities, along with low water solubility in the acidic pH range. It also showed low foaming capacity and stability. The average of the sensory and acceptability scores for the burger formulations were 6.6 on a 9.0-point hedonic scale, indicating that cashew nut protein concentrate can replace soy protein in these formulations.

The protein concentration in alternative products can be obtained via different techniques used to isolate proteins from food. These concentrates have wide application in the food industry due to their important technological properties, such as the determination of solubility, fat binding and retention capacity, water retention, emulsification, viscosity, thermal stability, gel formation, and foaming ability [78].

Lima et al. [77] evaluated plant-based burgers with ingredients such as cashew fiber, cowpea puree, tomato, onion, chili, garlic, black pepper, dehydrated parsley, salt, and corn oil. This product, characterized by low fat content and low energy, presented promising sensory properties as well. The composition indicated 71.08% moisture, 2.07% ash, 4.86% proteins, 1.19% lipids, and 20.79% total carbohydrates.

#### 3.2.4. Burgers Made with Algae

Algae are considered a little-known source of protein in Europe but are a traditional ingredient in Asian cuisine. They have great potential for creating new products such as meat analogs. Algae have a significant proportion of necessary amino acids and synthesize long-chain polyunsaturated fatty acids, which are beneficial for human health, while also containing colors, vitamins, and fibers [27,60,79].

Algae (*Arthrospira platensis*, *Isochrysis, Picochlorum*, and *Chlorella*) have become ingredients to produce new food products with functional and nutritional quality [47]. Fu et al. [80] mentioned that algae are an excellent source of protein for meat analogs, especially *Chlorella vulgaris*, in which the methionine amino acid content is compared to that of beef. As noted by Atitallah and Barkallah et al. [47], algae can also be a suitable source of beneficial natural flavorings and colorings.

Fernandes et al. [32] evaluated high nutritional value plant-based burgers, incorporating algae, peas, and chickpeas, and the following results were obtained: protein (8.01%), sugar content (1.71%), fiber (5.75%), and total fat (5.35%). A shelf-life analysis confirmed that the plant-based burger with algae is safe to be consumed for at least 90 days when stored in its original packaging and refrigerated [32]. Table 3 demonstrates the development of meat analog burgers associated with fruits, legumes, cereals, algae, nuts, and mushrooms, without the addition of animal meat.

### 3.3. Mushrooms in Burgers

The growing popularity of alternative foods, such as meat analog options, reflects a conscious shift in food choices and in contemporary cuisine. In this context, new burger options that feature edible mushrooms as the main ingredient stand out, an innovative proposal discussed by Perez-Montes et al. [30] and Patinho et al. [31], who emphasized the relevance of this approach, highlighting the mushroom not only for its nutritional value and antioxidant properties but also for contributing to the unique texture and taste of these products.

Perez-Montes et al. [30] reported that mushrooms are products with remarkable taste and aromas, in addition to bioactive compounds and nutritional values. In general, mushrooms are rich in proteins, carbohydrates, fibers, and essential amino acids. Sayeed Ibrahim and Huda-Faujan [58] described mushrooms as a promising source of bioactive compounds such as phenolics (3–11 mg/g) and flavonoids (2.5–4.8 mg/g); both authors support the use of this product as an ingredient in hybrid burgers as well.

## 4. Final Considerations

The physicochemical and sensory characterization of these new hybrid products and plant-based burgers demonstrated the diversity of ingredients, as well as their possible associations in burgers. When considering the analysis performed in these studies, it can be inferred that this study provided detailed information on the composition, processing, and sensory characteristics of meat analogs, thus contributing to an in-depth understanding of the advancements already made and the challenges still faced.

This study also highlighted the growth of the plant-based protein market and the change in dietary patterns, reflecting a growing concern about cost, safety, and sustainability. However, despite the market potential of these products, there are still challenges, such as texture and taste, which are important characteristics of animal-meat burgers. Another point to consider is the diversity of preferences among consumers with different beliefs or eating styles. Vegans, for example, may not prefer a product that is so close in taste or texture to meat, unlike flexitarians who are looking for a product that is similar in all attributes to animal meat. The development of formulations that take into account the preferences of consumers from groups that follow different dietary patterns is therefore recommended. Although there are already several options on the market, these characteristics are still limited.

We reiterate that, nutritionally, the combination of different food groups, such as legumes + cereals, can be promising for increasing the quality of the amino acid content of products. Additionallyl, creative and innovative formulations, chemical and technological quality, and consumer acceptability all play very important roles in the growth of this market.

This study also highlighted the ongoing need for research in and development of these new products, whether they are plant-based or hybrid products, using ingredients while simultaneously reducing waste, minimize the impact of environmental footprints, and thus expanding on the available options through improvements.

## Figures and Tables

**Figure 1 foods-13-03855-f001:**
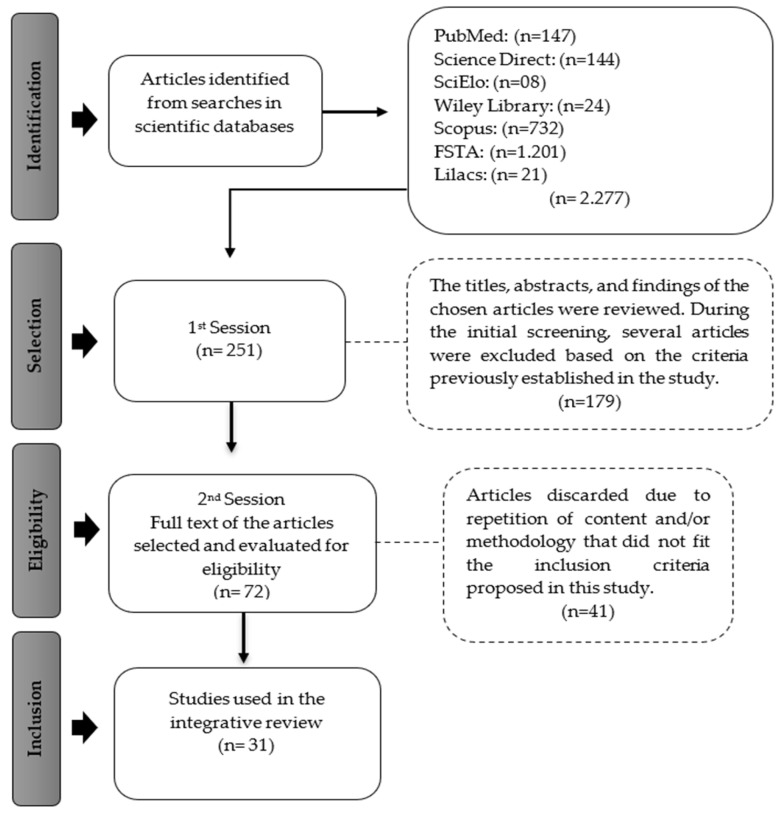
Flowchart of the scientific dataset. Source: Study data.

**Figure 2 foods-13-03855-f002:**
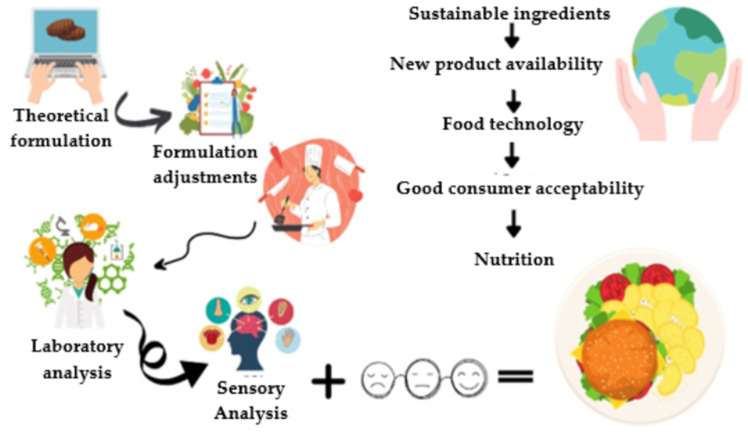
Relevant factors in new product development. Source: Authorship by the authors.

**Table 1 foods-13-03855-t001:** Findings of hybrid burgers in combination with animal meat and other proteins.

Meat + Options Available	Ingredients	Authors	Country
Cereals and pseudocereals	Soy protein and bread crumb with quinoa and buckwheat flour in functional beef burger formulation.	Bahmanyar et al. [53].	Iran
Legumes	Hybrid meat burgers made with pulses and chicken. Hybrid meat and plant-based burger patty. Yellow pea and ground chicken hybrid meat burgers.	Chandler & McSweeney [17]; Petrat-Melin & Dam [1]; Barker; McSweeney [49].	Canada Denmark Nova Scotia, Canada
Nuts	Lamb burgers, using partially or totally lamb meat, lamb fat, and thickener (corn starch), with oils and flours from different seeds and nuts. Beef burgers using a tiger nut (*Cyperus esculentus* L.) oil emulsion.	Rabadán et al. [54]; Carvalho Barros et al. [55].	Spain Spain
Vegetables	Beef, plant-based, and hybrid beef burgers. Hybrid meat products with pork, cattle, chicken, and vegetables. Beef burgers with added non-conventional food plants. Chicken burgers blended with green banana and passion fruit epicarp biomasses.	Grasso et al. [43]; Grasso & Goksen [41]; Ziegler et al. [46]. Santos et al. [52].	United Kingdom Ireland Brazil Brazil
Fruits			
Edible mushrooms	Hybrid beef and plant-based burgers. Mushroom stems in hybrid products and meat analogs.	Sogari et al. [56]; Sayeed Ibrahim; Huda-Faujan [57].	Italy and USA Malaysia
Algae	Seaweed powder (*Undaria* sp.) as a functional ingredient in low-fat pork burgers.	Nagai et al. [58];	Argentina and Spain

**Table 2 foods-13-03855-t002:** Data on hybrid and plant-based burgers, including nutritional information, sensory characteristics, and an overview of different protein sources.

Type of Burgers/Food Ingredients	Formulation/ Processing	Nutritional Aspects	Sensory Analysis	Findings	Authors/ Country
Hybrid (lean ground chicken, yellow peas, chickpeas, lentils)	For the control burger, 495 g of ground chicken and 5 g of salt were used. In the burgers with legumes, peas, chickpeas, and lentils, in the proportions of 25%, 50%, and 75%, replaced the same amount of ground chicken.	Scores for yellow pea, chickpeas, lentils, and the control (50%), respectively: protein—14.06, 18.03, 19.56, and 17.38%; fat—10.87, 10.21, 9.98, and 11.91%; ash—9.98, 1.52, 1.27, and 1.66%; moisture—61.03, 59.87, 61.31, and 70.01%.	--	This study demonstrated that legumes can be combined with chicken to make new product; however, the authors believe in low substitution amounts (25% level).	Chandler; McSweeney [17]. Canada
Hybrid (beef, soy protein)	The protein component in terms of the 100% beef to 100% plant-based ground beef ratios were 100:0, 75:25, 50:50, 25:75, and 0:100.	Animal-meat content increased the scores for *saltiness* and *flavor intensity*, indicating that plant-based meat analogs require a higher salt content to achieve similar levels of salt and flavor perception.	Although the samples were perceived as having clearly different aroma and taste attributes, their relative aroma and taste intensities were similar across the set.	The 75:25 hybrid burger (beef: plant-based) resembles the 100:0 burger. The 50:50 burgers (beef: plant-based) are supposedly the least oily samples.	Chin et al. [60]. Australia
Hybrid (beef and plan-based protein)	50:50% beef: plant-based (2/3 pea protein; 1/3 fava bean and ground pea starch) burger.	--	In the CATA test, the descriptor “juicy” was more often used to describe the hybrid than the beef burger (53% vs. 12%).	The hybrid burger, made with 50% plant-based ingredients and 50% meat can be a viable alternative to the traditional burger, combining the nutritional and environmental benefits of plants with the sensory characteristics of meat.	Petrat-Melin; Dam [1]. Denmark.
Plant-based (porang and tempeh)	The proportions of porang: tempeh were 75:25, 50:50, and 25:75 compared to the control (100% tempeh).	Scores for 25:75 porang: tempeh; moisture—52.00; ash—0.49; protein—17.75; fat—4.17; carbohydrates—25.57.	--	The porang and tempeh ratio of 25:75 showed lower cooking loss and higher yield compared to the control.	Herawati et al. [29]. Indonesia
Plant-based (white bean *tempeh* and soybean *tempeh*)	To produce tempeh, 200 g of beans was hydrated in water with vinegar (5%) for 20 h and autoclaved at 121 °C for 15 min.	Scores for white bean *tempeh* and soybean *tempeh*: moisture—3.94 and 2.57%; ash—2.40 and 2.03 g/100 g; protein—23.34 and 43.74 g/100 g; fat—1.29 and 24.88 g/100 g; Kcal—326.77 and 440.44 g/100 g.	Scores for white bean and soybean *tempeh*: appearance—6.22 and 6.93; aroma—4.00 and 6.54; taste—3.55 and 6.35; overall acceptance—5.10 and 6.40.	The white bean tempeh burgers had similar crispiness, appearance, and consistency but received a lower taste score than the soy ones, possibly due to the bean aftertaste.	Vital et al. [76]. Brazil.
Plant-based (quinoa, corn, and lupins)	Quinoa and corn were pressure-cooked for 19 and 40 min, respectively; lupin was ground and mixed.	Scores for quinoa, corn, and lupin, respectively: protein—17.63, 8.15, and 61.44%; fat—6.76, 4.58, and 18.29%; carbohydrates—73.03, 85.08, and 17.95%.	Soft texture, nice taste, and light brown color.	The use of Andean crops favored the protein content and the contribution of sulfur amino acids and tryptophan from quinoa and lysine and threonine from lupin.	Chilon-Llico et al. [25]. Peru.
Plant-based (cashew nuts and soy protein)	Treated cashew fiber (27.0%), medium-sized light-textured soy protein (27.0%).	Scores for the soy protein burger and cashew nut concentrate: protein—6.7 and 7.1%; fat—1.3 and 1.5%; Kcal—118.0 and 115.4/100 g.	Overall acceptance was 6.6 on a 9-point scale.	The concentrate obtained showed high water absorption capacity and oil absorption capacity and low water solubility.	Lima et al. [28]. Brazil.
Plant-based (cashew fiber and cowpea puree)	29.3% cashew fiber and 25.1% cowpea paste.	Moisture—71.08%; proteins—4.86%; fat—1.19%; carbohydrates—20.79%.	Overall acceptance was 7.8 on a 9-point scale.	Low fat and calories, and good stability during storage.	Lima et al. [77]. Brazil.

**Table 3 foods-13-03855-t003:** Findings of plant-based burgers made with plant proteins, mushrooms, and their combinations.

Plant-Based Burgers			
Meat + Options Available	Ingredients	Author	Country
Cereals and pseudocereals	Pre-cooked, legume-based burgers, pearled barley (*Hordeum vulgare* L.), and pre-cooked corn (*Zea mays* L.). Burgers made with quinoa, lupin, and corn.	Summo et al. [75]; Chilón-Llico et al. [25].	Italy Peru
Legumes	Beef-flavor burgers with pea proteins, lentils, and faba beans. Soy burgers and pea–faba burgers. Enriched with pea protein texturing as a substitute for meat in hamburgers.	Kim et al. [26]; Sousa et al. [13]; Peñaranda et al. [27].	USA Switzerland Spain
Nuts	Cashew kernel protein concentrate from nut processing by-product and its use in plant-based burgers.	Lima et al. [28].	Brazil
Vegetables	Meat analog burgers with pea protein enriched with lucerne, spinach, and chlorella. Vegetable cashew burger with broken cashew kernels. Porang with tempeh in burgers.	Peñaranda et al. [27]; Lima et al. [28]; Herawati, Kamsiati e Widowati [29].	Spain Brazil Indonesia
Fruits	Plant-based burgers made with cashew juice residue (fiber) mixed with cowpea (protein source). Jackfruit burgers made vegan with cabbage flour.	Lima et al. [77]; Lima Segundo et al. [81].	Brazil Brazil
Edible mushrooms	The addition of mushrooms into meat products. Burgers with mushrooms.	Perez-Montes et al. [30]; Patinho et al. [31].	Mexico Brazil
Algae	Burgers suitable for vegan and vegetarian diets using grass pea and seaweed as the main ingredients. Meat analog burgers with pea protein enriched with lucerne, spinach, and chlorella. Microalgae as a sustainable ingredient for meat analogs.	Fernandes et al. [32]; Peñaranda et al. [27]; Zhu et al. [40].	Portugal Spain China

## Data Availability

No new data were created or analyzed in this study.

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
