# Peer review of "Hybrid and Plant-Based Burgers: Trends, Challenges, and Physicochemical and Sensory Qualities"

_foods, 2024, doi:10.3390/foods13233855_

Round 1

Reviewer 1 Report

Comments and Suggestions for Authors

The present literature review by Merces et al. summarizes the recent research in the field of hybrid burgers. The language is easy to understand and clear. The hypothesis is clear and sound. My suggestions are as follows-

        i.            Abstract: L19-25 seems too general and needs to add salient findings/ conclusions and recommendations

      ii.            Keywords: meat analogue and meat alternatives- please keep any one of them and add other relevant keywords

    iii.            L31-32: please dd recent data if available

    iv.            L 40: these new alternatives are substitutes: please rephrase it for a better understanding

      v.            Heading 3 need to restructuring/ rearranging to get better clarity and continuity. Some information in this section may be summarized in the tabular form also.

    vi.             3.1.1: not clear;

  vii.            L232: scientific name in italics

viii.            Heading 3.2: may be plant proteins used in burger preparation and then in sub-heading, describe the various sources; The cereals may be grouped under a separate sub-heading;

Also, please mention the challenges in utilization of each protein if available in the literature.

    ix.            Heading 4: may add more content highlighting the future trend, prospects and challenges.

      x.            Figures excellent

Author Response

Dear Editor and Referee,

We would like to thank the referee for your precious time during the paper evaluation and the useful suggestions that helped us to improve the quality of the manuscript. We are grateful for all the feedback on our paper and agree with all of them.           

We have added the suggestions and corrections pointed out by the referee. Questions and suggestions are in black and answers are in blue. We hope that we have now accomplished all the corrections requested by our reviewers.

 Comments to Author:

# Reviewer 1

Comments and Suggestions for Authors

The present literature review by Merces et al. summarizes the recent research in the field of hybrid burgers. The language is easy to understand and clear. The hypothesis is clear and sound. My suggestions are as follows.

Response: Thanks for your feedback, we have given our best on this paper and we really believe that this review will be useful for research fellows and students. All suggestions of our reviewer were accepted, we understand when other people read it, they may have a different perception and we believe that more people reviewing it and giving suggestions will improve the   paper quality.

  1. Abstract: L19-25 seems too general and needs to add salient findings/ conclusions and recommendations

Response: We agree with your request and we have inserted new paragraphs and more authors about the part you requested.

  1. Keywords: meat analogue and meat alternatives- please keep any one of them and add other relevant keywords

Response: We agree to your suggestion and have inserted new word in the abstract, we hope you like with this new version.

iii.L31-32: please dd recent data if available

Response: It was corrected as the referee has suggested.

  1. L 40: these new alternatives are substitutes: please rephrase it for a better understanding

Response: We have improved it in this new version as proposed by our referee.

  1. Heading 3 need to restructuring/ rearranging to get better clarity and continuity. Some information in this section may be summarized in the tabular form also.

Response: We have improved it in this new version as proposed by our referee.

    vi 3.1.1: not clear;

Response: The sentence was amended; we hope you enjoy this new version. We agree it is better than the previous one.

  vii. L232: scientific name in italics

Response:  Dear reviewer, thank you for your attentive remarks, we have worked with focus on this section of our manuscript to make it clearer. We hope that this version is suitable.

viii. Heading 3.2: may be plant proteins used in burger preparation and then in sub-heading, describe the various sources; The cereals may be grouped under a separate sub-heading; Also, please mention the challenges in utilization of each protein if available in the literature.

Response: We reorganized the groups as suggested by our referee, but sometimes they are together in the same formulation. Thank you for this suggestion, it is a very good one, by the way.  We recognize the relevance of this idea.

  1. Heading 4: may add more content highlighting the future trend, prospects and challenges.

Response: We followed our referee’s suggestion.

  1. Figures excellent

Response: Thanks for your inspiring comments and advices, we have worked hard on this project/paper and we really believe that this review will be helpful for students and researchers.

We are very grateful,

The authors

Reviewer 2 Report

Comments and Suggestions for Authors

Dear Authors,

My comments are attached.

Comments on the Quality of English Language

The English language needs some editing.

Author Response

Dear Editor and Referee,

We would like to thank the referee for your precious time during the paper evaluation and the useful suggestions that helped us to improve the quality of the manuscript. We are grateful for all the feedback on our paper and agree with all of them.     

 We have added the suggestions and corrections pointed out by the referee. Questions and suggestions are in black and answers are in blue. We hope that we have now accomplished all the corrections requested by our reviewers.

# Reviewer 2

Hybrid and meat analogues burgers: Trends, challenges, physico-chemical and sensory quality of these products

This review provides an extensive overview of hybrid and plant-based burgers, covering their formulation, sensory attributes, and market potential. However, some areas could be improved regarding clarity, structure, and conciseness.  Additionally, English needs to be improved. Many sentences must be rephrased for conciseness and clarity.

Response: Thanks for your feedback, we have given our best on this paper and we really believe that this review will be useful for research fellows and students. All suggestions of our reviewer were accepted, we understand when other people read it, they may have a different perception and we believe that more people reviewing it and giving suggestions will improve the paper quality.

Title:

 Please delete “of these products”.

Response: The terms were deleted in the title as referee’s suggestion.

 Consider adding “nutritional quality” 

Abstract: We usually prefer “nutritional” as well, we even have nutritionists in our research group, but since we are also working with phenolic compounds, bioactive compounds, pH, we thought it best to keep “chemical” because it covers more of the approaches we made in the paper. Thank you for your care with our paper.

 The abstract should more clearly differentiate between hybrid burgers and entirely plant based burgers. Please rewrite this part.

Response: We have improved it in this new version as proposed by our referee.

 Please specify which databases were searched and clarify what "specific descriptors" refer  to.

Response: We followed our referee’s suggestion.

 The description of "physicochemical and sensory quality" could be clearer (e.g., add

texture, taste, nutritional profile, etc.).

Response: We have tried to make it clearer in this new version of the manuscript. We think it's really better with your suggestion.

 Please specify "technological innovations" (Line 15).

Introduction: We improved the paragraph as suggested by our referee

 Please define "hybrid burgers" and "meat-analogue burgers" earlier in the text and clearly distinguish between them.

Response: It was improved as the referee has suggested.

 Please specify the novelty or unique angle of this study.

Response: We added this information as suggested by our referee

 Please use consistent terminology throughout the text instead of shifting between "plant based protein" and "plant meat analogues."

Response:  We thank our reviewer for your suggestion. We agree that it looks better the way you have recommended. It was rephrased. With your directions we understand what you meant.  We'll have to keep both terms, we apologize to our reviewer. For example, at times Plant-based is for products or food more general, while meat analogues really refer to hamburgers, meat balls, nuggets...

Materials and Methods:

 Briefly explain why English, Portuguese, and Spanish were chosen for the review.

Response: We reorganized this paragraph according to your suggestions. Previously we have listed these languages because they are the only ones we are fluent in and we don't need a language translator. Sorry, but we've tried using other languages with a language translator in a previous study and the quality wasn't very good. We also believe that English, at least in the abstract, can capture the great majority of papers in the databases.

 Please list the “exclusion criteria”.

Response: We have improved it in this new version as proposed by our referee.

Results and Discussion:

 The connection between the reported findings and the study’s purpose is generally unclear. Explain how each finding contributes to understanding trends, technological innovations,  or consumer acceptance in the context of hybrid and plant-based burgers.

 Response: Thank you for your comments and recommendations. We adjusted the paragraphs, improved them with more discussions that will be in blue in the text. 

 Please provide more details on consumer preference – e.g., why flexitarians prefer hybrid  products and why omnivores may hesitate to try plant-based burgers, possibly due to  texture, flavour, or familiarity concerns.

Response: We thank the referee for the gentle and encouragement words and new ideas.

 Expand on the discussion of consumer diversity by addressing how different demographic or dietary preferences might shape future product development.

 Response: We thank our reviewer for your suggestion. We agree that it looks better the way you have recommended.

 The whole section lacks a cohesive narrative connecting the research findings with relevant literature. Please link these findings to the study’s objectives. Please connect the relevance of each study (found in the database) with the broader trends and mention recommendations for future research.

Response: It was improved as the referee has suggested. Thanks for your inspiring comments and advices, we have worked hard on this project/paper and we really believe that this review will be helpful for students and researchers.

Conclusion:

 Phrases like “possibilities for combinations in the production of burger” (Lines 353-354) and “the evolution of this dynamic market” (Line 363) must be rephrased to sound more concise.

Response: We agree with your request and we have inserted new paragraphs and more authors about the part you requested.

 Please provide specific recommendations for future research or industry focus.

Response: We strongly agree. Thanks for your suggestion

We are very grateful,

The authors

Reviewer 3 Report

Comments and Suggestions for Authors

This review provides a timely and relevant overview of hybrid and plant-based burger alternatives, addressing an important topic in food science and sustainable food production.  The manuscript is generally well-structured but requires some refinements to enhance its scientific rigor and clarity.

1. Title lacks specificity about the review's scope (2017-2024) and methodology (integrative review).

2. Abstract line 19-21: Statement "The physicochemical and sensory quality...demonstrated the variety of ingredients" is vague and needs quantifiable results.

3. Abstract lacks specific findings from the review (e.g., number of studies analyzed, key trends identified).

4. Lines 31-32: Market size data from 2020 ($862 billion) is outdated.   Need 2023/2024 data.

5. Lines 37-39: The 68% growth figure for European plant-based sector needs context (baseline values).

6. Lines 40-46: Too many alternative terms listed without clarifying preferred terminology.

7. Line 46: Statement about "sustainable and healthy food choices" needs supporting evidence.

8. There is a gap in the literature coverage on innovative processing technologies and emerging protein sources. You may refer to the following literature: https://doi.org/10.1016/j.foodchem.2024.139360.

9. Methodology Critical Flaws:

- Figure 1 flowchart:

* Missing initial number of records identified

* No explanation of screening process

* Lacks number of full-text articles assessed

* Missing reasons for exclusion numbers

- Search Strategy:

* Limited database selection (PubMed, Science Direct, SciElo, Wiley)

* Missing important databases (Web of Science, Scopus)

* No Boolean operators or search strings provided

* No hand-searching of reference lists mentioned

10. Hybrid Burgers Section (Lines 92-136):

- Lacks systematic organization of findings

- Missing comparison table of different hybrid burger formulations

- No clear synthesis of optimal meat-to-plant ratios

- Specific data gaps: Processing parameters, Cost comparisons, Shelf-life data, and Nutritional composition tables.

11. Plant-based Section:

- Lines 207-228: Discussion of legumes is superficial

- Missing critical analysis of: Protein quality comparisons, Functional properties, Processing requirements, and Cost implications.

12. Technical Content Gaps: No discussion of: Water holding capacity, Binding mechanisms, Texture profile analysis data, Storage stability, Packaging requirements, and Food safety considerations.

13. Lines 107-110: "The addition of these plants flour improved nutritional properties". Need specific values: Protein content changes, Fiber content increases, Fat reduction percentages.

14. Statistical Analysis Issues:

- No meta-analysis of comparable studies

- Missing effect sizes

- Lack of statistical comparison between different formulations

- No confidence intervals reported

Author Response

Dear Editor and Referee,

We would like to thank the referee for your precious time during the paper evaluation and the useful suggestions that helped us to improve the quality of the manuscript. We are grateful for all the feedback on our paper and agree with all of them.           

We have added the suggestions and corrections pointed out by the referee. Questions and suggestions are in black and answers are in blue. We hope that we have now accomplished all the corrections requested by our reviewers.

# Reviewer 3

Comments and Suggestions for Authors

This review provides a timely and relevant overview of hybrid and plant-based burger alternatives, addressing an important topic in food science and sustainable food production.  The manuscript is generally well-structured but requires some refinements to enhance its scientific rigor and clarity.

 Response: Thank you for your motivational comments and recommendations, we have worked focused on your suggestions and we hope that this version is appropriate. We adjusted the paragraphs, improved them with more discussions that will be in blue in the text. 

  1. Title lacks specificity about the review's scope (2017-2024) and methodology (integrative review).

Response: Thank you for the advice. We have improved it in this new version as proposed by our referee.

  1. Abstract line 19-21: Statement "The physicochemical and sensory quality...demonstrated the variety of ingredients" is vague and needs quantifiable results.

Response: We have improved it in this new version as proposed by our referee.

  1. Abstract lacks specific findings from the review (e.g., number of studies analyzed, key trends identified).

Response: We are very grateful for the kindness of our reviewer. We have rewritten the abstract again to qualify it. Thank you for your suggestion.

  1. Lines 31-32: Market size data from 2020 ($862 billion) is outdated. Need 2023/2024 data.

Response:  Thank you for your attentive remarks, we have worked with focus on this new version of the document. We hope that this one is suitable.

  1. Lines 37-39: The 68% growth figure for European plant-based sector needs context (baseline values).

Response: You are right, it was written in a poor way. We have improved and clarified it throughout the paper to avoid ambiguity. Thank you for helping.

  1. Lines 40-46: Too many alternative terms listed without clarifying preferred terminology.

Response: It was improved, it was really weak, we hope it is clearer at this time. It was amended, we hope you enjoy this new version.

  1. Line 46: Statement about "sustainable and healthy food choices" needs supporting evidence.

Response:  Thank you for your attentive remarks, we believe that with your help this part of our manuscript is clearer.

  1. There is a gap in the literature coverage on innovative processing technologies and emerging protein sources. You may refer to the following literature: https://doi.org/10.1016/j.foodchem.2024.139360.

Response: We have included this paper in this new version as proposed by our referee. Thank you for your gentle suggestion.

  1. Methodology Critical Flaws:

- Figure 1 flowchart:

* Missing initial number of records identified

* No explanation of screening process

* Lacks number of full-text articles assessed

* Missing reasons for exclusion numbers

Response: We have improved it in this new version as proposed by our referee. Thank you for your suggestions.

- Search Strategy:

* Limited database selection (PubMed, Science Direct, SciElo, Wiley)

* Missing important databases (Web of Science, Scopus)

* No Boolean operators or search strings provided

* No hand-searching of reference lists mentioned

 Response: We have tried to make  this section clearer in this new version of the manuscript. We think it's really better with your suggestion.

  1. Hybrid Burgers Section (Lines 92-136):

- Lacks systematic organization of findings

- Missing comparison table of different hybrid burger formulations

- No clear synthesis of optimal meat-to-plant ratios

- Specific data gaps: Processing parameters, Cost comparisons, Shelf-life data, and Nutritional composition tables.

Response: It was rephrased. With your directions we understand what you meant. If something is not exactly as it was suggested, please let us know and we can improve it again.

  1. Plant-based Section:

- Lines 207-228: Discussion of legumes is superficial

- Missing critical analysis of: Protein quality comparisons, Functional properties, Processing requirements, and Cost implications.

 Response: The section was amended; we hope you enjoy this new version. We agree it is better than the previous one.

  1. Technical Content Gaps: No discussion of: Water holding capacity, Binding mechanisms, Texture profile analysis data, Storage stability, Packaging requirements, and Food safety considerations.

Response:  We thank our referee for the helpful suggestions. Everything that our referee requested, we added, and we agree that this version is much more complete with these suggestions.

  1. Lines 107-110: "The addition of these plants flour improved nutritional properties". Need specific values: Protein content changes, Fiber content increases, Fat reduction percentages.

Response:   You are completely right. We apologize for our lack of attention, and  we have tried to  improve all of them.

  1. Statistical Analysis Issues:

- No meta-analysis of comparable studies

- Missing effect sizes

- Lack of statistical comparison between different formulations

- No confidence intervals reported

Response: It was corrected as the referee has suggested.  We considered and followed our referee's suggestion. We hope you agree with this new version. Only, statistical analysis was not performed, since it is not a systematic review nor meta-analysis.

We are very grateful,

The authors

Round 2

Reviewer 2 Report

Comments and Suggestions for Authors

Dear Authors, 

Thank you for the revised manuscript. I have only one additional comment: Please change "hamburgers" to "burgers" in the title and anywhere else in the text. The term "hamburgers" refers explicitly to burgers made from beef.

Author Response

Dear Referee 2,

              We would like to thank our reviewer again for your valuable time during this 2º review and all your insightful comments which helped us to improve the quality of the manuscript.

We really have appreciated all the queries and we agreed with all of them because we believe it will improve the final quality of the paper.                                                                                   

 We have adjsted "burgers" as you pointed out.  They are all in brown this time.

Best wishes,   The authors

Reviewer 3 Report

Comments and Suggestions for Authors

All my comments have been properly addressed.

Author Response

Dear Referee,

   We would like to thank our reviewer again for your valuable time during the review process and the insightful comments which helped us to improve the quality of the manuscript.

We really have appreciated all the queries and we agreed with all of them because we believe it will improve the final quality of the paper.         

Best wishes, The authors